# Clinical Challenges of Emerging and Re-Emerging Yeast Infections in the Context of the COVID-19 Pandemic

**DOI:** 10.3390/microorganisms10112223

**Published:** 2022-11-10

**Authors:** Dario Corrêa-Junior, Iara Bastos de Andrade, Vinicius Alves, Glauber R. de S. Araújo, Susana Frases

**Affiliations:** 1Laboratório de Biofísica de Fungos, Instituto de Biofísica Carlos Chagas Filho, Universidade Federal do Rio de Janeiro, Cidade Universitária, Ilha do Fundão, Rio de Janeiro, CEP 21941-902, Brazil; 2Rede Micologia RJ, FAPERJ, Rio de Janeiro, CEP 21941-902, Brazil

**Keywords:** COVID-19, fungal co-infection, immunomodulation

## Abstract

During the geological eras, some fungi, through adaptation and/or environmental/ecological pressure, interacted directly and indirectly with humans, through occasionally harmful interaction interdependent on the individual’s immunological condition. Infections caused by yeasts are underreported, subjugated, and underdiagnosed, and treatment is restricted to a few drugs, even after the significant progress of medicine and pharmacology. In the last centuries, antagonistically, there has been an exponential increase of immunocompromised individuals due to the use of immunosuppressive drugs such as corticosteroids, increased cases of transplants, chemotherapeutics, autoimmune diseases, neoplasms, and, more recently, coronavirus disease 2019 (COVID-19). This review aims to survey emerging and re-emerging yeast infections in the current clinical context. Currently, there is an immense clinical challenge for the rapid and correct diagnosis and treatment of systemic mycoses caused by yeasts due to the terrible increase in cases in the current context of COVID-19.

## 1. Introduction

The kingdom “Fungi” comprises from 1.5 to 5 million species, of which around 600,000 have already been described. A few hundred cause diseases in humans, and a few dozen can affect healthy people [1]. Fungal pathogens probably suffered from environmental pressure and, over time, may have adapted their pathogenic repertoire primarily to fewer complex organisms such as plants, environmental predators, invertebrates, and lower mammals—before encountering and interacting with humans [2,3]. Several fungal pathogens are environmental microorganisms for which the human being is an accidental host; on the other hand, there are several fungi that are commensals that can emerge due to an imbalance of the immune system [4].

For a fungus to cause damage to a human and thus cause a certain disease, this microorganism needs, at least, a triad of mechanisms: (I) to have the ability to obtain nutrients to maintain its metabolism, (II) to be able to grow at temperatures above 36 °C, and (III) being able to circumvent, evade or hoodwink the immune system [5].

Serious fungal diseases rarely occur in healthy individuals; they are more frequent in prodromal individuals, with underlying disease and/or serious diseases. The advancement of contemporary medicine often makes its main beneficiaries vulnerable to fungal infections, constituting a poor prognosis for different clinical conditions, such as neoplasms, solid (allogeneic) organ and stem cell transplants, gene therapies, therapies for autoimmune diseases, procedures surgical procedures, implantation of devices (stent, pacemaker, valves, artificial joints, etc.), individuals living with HIV/AIDS, and, more recently, patients with SARS-CoV-2 [6,7].

In the last decades, there has been an exponential growth in the knowledge of human fungal pathogens—due to the development or improvement of specific genetic, genomic, and molecular tools for these microorganisms—but a lethargic development of drugs and a technical limitation in the diagnosis and accurate statistics of the incidence of these fungal pathogens, because the data of cases of fungal infection are underestimated [8]. In addition, fungi are adapting to climate change (e.g., increase in temperature) and environmental pressures (agricultural pesticides). Fungi that were previously not a threat to human health because their mechanisms did not satisfy the conditions of the triad, are becoming increasingly common in infection of humans; in many cases, these fungi are resistant to antifungals recommended for treatment [9,10,11,12].

## 2. Influence of Drugs on Host Defense: Corticosteroids (Glucocorticoids, GCs)

The multiple therapeutic options for patients with malignant diseases—with emphasis on the various chemotherapy regimens and the growing use of organs for transplants or hematopoietic cell transplants—and the increased survival of patients with autoimmune diseases, together, have greatly increased the number of immunosuppressed hosts. Such patients have been characterized by their susceptibility to fungal infections. The use of immunosuppressive drugs is an advance in the treatment of several diseases in which interleukins and cytokines play a pathophysiological role. Among these drugs, glucocorticoids have acquired clinical relevance.

Corticosteroids affect virtually every tissue in the body, including immune cells, mitigating migration and recruitment [13]. The term “corticosteroids” is broad, encompassing all steroid hormones. Both endogenous and exogenous corticosteroids cause immunosuppression and increase the incidence of infectious diseases. Glucocorticosteroids (GCs) are associated with an increased incidence of opportunistic infection, worsening infection severity, and recrudescence of latent viral infections [14].

GC production is orchestrated by the circadian cycle and regulated by the hypothalamic–pituitary–adrenal (HPA) axis. Inputs from the suprachiasmatic nucleus (SCN) stimulate the paraventricular nucleus (PVN) of the hypothalamus to release corticotropin-releasing hormone (CRH) and arginine vasopressin (AVP). These hormones act on the anterior pituitary, where they activate corticotropic cells to secrete adrenocorticotropin hormone (ACTH) into the general circulation. Subsequently, ACTH acts on the adrenal cortex to stimulate the synthesis and release of glucocorticoids [15].

Glucocorticoids, including endogenous cortisol, mainly exhibit anti-inflammatory and immunosuppressive properties and are widely used in the treatment of various inflammatory, immune-mediated, and neoplastic diseases of animals and humans, and their benefits and harms have been known for over seven decades [15,16]. The main organs responsible for the synthesis of glucocorticoids are the suprarenal glands and adrenal endocrine glands. However, there is evidence that other organs, such as the skin, intestine, and heart, are capable of synthesizing glucocorticoids, since the presence of steroidogenic enzymes has already been detected, as well as significant levels of GCs, after adrenalectomy [17]. The physiological and pharmacological functions of glucocorticoids are mediated by the intracellular glucocorticoid receptor, ubiquitous throughout the body, but with heterogeneity in response [15].

GCs inhibit the activation, proliferation, and survival of T-lymphocytes, as well as several cytokines derived from this cell type. They also promotes a change in the immune response profile from Th1 to Th2 at physiological concentrations [18]. At supraphysiological doses, they reduce the activity of the STAT4 transcription factor and, consequently, the release of Th1 cytokines, including interleukin (IL)-2, interferon (INF)-γ, IL-4, IL-5, and IL-13, and reduce the activity of the transcription factor GATA-3 [19]. There is influence on the humoral immune response through apoptosis and anergy of B lymphocytes [20]. In the innate response, GCs inhibit the release of cytokines and inflammatory mediators, including lipid mediators and reactive oxygen species by macrophages [21].

## 3. COVID-19 in the Perspective of Immunosuppression

The infection by coronavirus disease 2019 (COVID-19), responsible for severe acute respiratory syndrome coronavirus 2 (SARS-CoV-2), possesses great variability in its immunohistopathological presentation; that is, it depends on the characteristics of the virus, the genetics, as well as the patient’s immunological condition. The disease may be asymptomatic, with mild/moderate symptoms, as it is in most cases; however, in some cases, it is fatal [22,23]. SARS-CoV-2 is a β-coronavirus, similar to coronaviruses that have been responsible for limited outbreaks in the last two decades, namely, severe acute respiratory syndrome coronavirus (SARS-CoV) and the Middle East respiratory syndrome-related coronavirus (MERS-CoV) [24,25].

SARS-CoV-2 infection can activate innate and adaptive immune responses and result in exacerbated inflammatory responses, including eosinopenia and lymphopenia, with severe reduction in the frequency of CD4+ and CD8+ T cells, B cells and natural killer (NK) cells. In addition, the infection can result in thrombocytopenia, with increased extracellular neutrophil traps and increased pyroptosis [26], increased levels of D-dimers [27], and coagulopathy, all of which are linked to a poor prognosis and may contribute to organ failure and death in patients with COVID-19 [28]. A worrying fact is that the tissue damage caused by the virus, which leads to excessive secretion of pro-inflammatory cytokines and the recruitment of other pro-inflammatory cells, such as granulocytes and macrophages [29,30], is the trigger of secondary hemophagocytic lymphohistiocytosis (sHLH) popularly called “cytokine storm”. The treatment of COVID-19 is multidisciplinary (e.g., immunologists, rheumatologists, neurologists, physiotherapists, and hematologists), heterogeneous, and dependent on the viral–host relationship, but always aims to avoid the cytokine storm and tissue damage, and to reduce morbidity and mortality.

Standard of care (SoC) is based on standard supportive care associated with different, and sometimes uncertain, pharmacological approaches, such as antivirals (e.g., Remdesivir, Lopinavir/Ritonavir), antibiotics (Azithromycin), immunomodulators (e.g., Tocilizumab) [31], convalescent plasma therapy [32], JAK inhibitors, IL-6 inhibitors, IL-1 inhibitors, intravenous immunoglobulin, or anti-TNF-α agents [26]. Furthermore, the World Health Organization strongly recommends the use of corticosteroids in the treatment of critically ill patients with COVID-19. The primary SoC is immunomodulation; vide that once immunological complications such as cytokine storm occur, antiviral treatment alone is not sufficient and must be combined with anti-inflammatory and immunosuppressive drugs. Although there are some unclear points about the modulation of the immune response using dexamethasone [33], this glucocorticoid, when used at 6 mg once daily for up to 10 days, reduced 28-day mortality in patients hospitalized with COVID-19 [34,35].

This essential approach to attenuating the immune system allows the patient, throughout the infection and after the infection by COVID-19, to be susceptible to opportunistic infections.

## 4. Emerging and Re-emerging Yeasts in the Context of Immunosuppressed Hosts

### 4.1. Non-Albicans Candida (NAC)

Among the fungi that cause diseases in humans, species of the genus *Candida* are among the most common. It is estimated that about 200 species of this genus have been described. *Candida* spp. can colonize several anatomical sites, such as skin, mucous membranes, respiratory tract, digestive tract, and urinary tract, and may even spread systemically through the host’s organs [36,37,38,39,40].

The main causative agent of candidiasis is *C. albicans*; however, in recent years, a tendency has been described to increase the incidence of mycoses caused by non-*albicans Candida* (NAC) species in humans, such as *C. auris, C. glabrata, C. dubliniensis, C. tropicalis, C. blankii, C. lusitaniae, C. tropicalis, C. krusei*, and *C. parapsilosis* [41,42,43,44].

Notable among the various factors that have been associated with the increase in the number of reported cases of these infections are the expansion in immunosuppressive therapies, irrational use of antifungals and broad-spectrum antibiotics, invasive surgical procedures, and comorbidities such as diabetes, hypertension, obesity, and cardiovascular and respiratory diseases [42,45,46].

*C. auris* has emerged globally as a multidrug-resistant healthcare-associated fungal pathogen [47] and was first reported isolated from the ear of a Japanese patient in 2009 [48]. This species is multiresistant to several antifungals available, such as fluconazole [49]. The emerging pathogen *C. auris* has been associated with nosocomial outbreaks on five continents [50,51,52,53,54,55]. The most common risk factors for *C. auris* infections are *diabetes mellitus*, advanced age, neutropenia, intensive care unit admission, lung disease, cardiovascular disease, kidney disease, interventions with medical devices such as catheters and mechanical ventilation, prolonged use of antibiotics, spectrum antifungals, and immunosuppressive therapy [56,57,58].

With the emergence of the COVID-19 pandemic, these factors associated with the increased incidence of fungal infections were intensified and NAC species are being identified as emerging new infections [59]. With the substantial increase in the number of hospitalizations for the treatment of respiratory symptoms caused by the SARS-CoV-2 virus, there has been an increase in the use of catheters and/or surgical procedures, which were already described as a means of entry for infections by *Candida* spp. [60]. As there was no effective drug treatment to combat COVID-19, several drugs were used in an uncontrolled manner, including antibiotics [61]. Finally, as the patients were hospitalized for a long time, the probability of contracting a fungal infection with species being largely resistant to the antifungal agents used in the human clinic increased [62].

During the pandemic, there has been an increasing number of reports of *C. auris* in COVID-19 acute care units. The researchers suspect that these outbreaks may be related to changes in routine infection control practices due to the health crisis. For example, there are limited availability of gloves and aprons, and poor cleaning and disinfection practices [63].

The irrational use of drugs has allowed many fungal species to present resistance to antifungal agents, making treatment impossible when necessary [45,46,60,61]. 

Patients with suspected or confirmed *C. auris* should be treated with echinocandins (caspofungin, micafungin, or anidulafungin), azoles (fluconazole, voriconazole, or itraconazole) and Amphotericin B and its liposomes. Monitoring of the therapeutic drug should be considered to optimize efficacy and limit azole toxicity [64]. As this is often a multidrug-resistant *Candida* infection, some strains may be resistant to all antifungals [65,66].

Another point to be highlighted is that deaths could be caused by the delay in the correct diagnosis—and, consequently, the beginning of an effective treatment—of infections by NAC species. Brikman and collaborators (2021) described a clinical case of a 67-year-old woman with a history of diabetes, hypertension, and ischemia, who, in addition to testing positive for SARS-CoV-2, began to show clinical signs of an infection. As a result, she started treatment with antibiotics (piperacillin/tazobactam and levofloxacin); however, it was only after some time that candidiasis caused by *C. parapsilosis* was diagnosed through blood culture. Even after starting caspofungin administration, the patient did not resist [67].

The clinical symptoms of candidemia are not specific and can often be confused with infections caused by other microorganisms. The laboratory diagnosis of an infection caused by emerging species of the genus *Candida* is essential for choosing the appropriate treatment; however, microbiological identification can be difficult since, depending on the time of infection, up to 50% of blood cultures can be negative [66]. The diagnosis of *Candida* spp. is through microscopic identification of fungal structures and complementary biochemical tests to differentiate some NAC species [68,69]. CHROMagar™ Candida agar medium (CHROMagar™, Paris, France) allows each species to grow a different color (*C. albicans*: blue-green, *C. tropicalis*: dark blue, *C. parapsilosis*: white, and *C. glabrata*: white, pink-purple), but as a disadvantage, this method does not cover all emerging NAC species [70]. Therefore, the most sensitive method for detecting non-*albicans Candida* infections is PCR, which can be done through the amplification of internal transcribed spacer 1 (ITS1), ITS2, and ITS4, and 18S, 5.8S, and 28S rRNA subunits; however, the need for an adequate infrastructure for PCR equipment, problems in sample preparation, and lack of standardization in PCR protocols is a disadvantage [42,71].

At the present time, post-COVID-19 clinical manifestations are emerging. Secondary hemophagocytic lymphohistiocytosis, adult multisystem inflammatory syndrome, and cytokine storm syndrome are hyperinflammatory episodes associated with SARS-CoV-2 and are being described as part of post-acute COVID-19 syndrome. Recently, Gautam and colleagues described a case of immune dysfunction related to hemophagocytic lymphohistiocytosis linked to steroid-induced immunosuppression that led to the patient developing disseminated *C. auris* infection, warning that post-acute COVID-19 syndrome puts patients at risk for opportunistic systemic mycoses. No consensus on guidelines and management of these patients has been determined so far. More studies of these cases are needed to formulate strategies for the diagnosis and management of these novel post-COVID-19 conditions. Meta-analysis studies showed an estimated global prevalence of 5.7% in patients with severe COVID-19 and mortality was estimated at 68% [72].

### 4.2. Cryptococcus *Spp.*

The *Cryptococcus* species complex mainly infects immunocompromised hosts, with the infection starting in the lungs with tropism for the central nervous system. The pathogenic species of *Cryptococcus* collectively cause over 200,000 infections of opportunistic cryptococcosis annually, placing them among the leading human pathogenic fungal species groups [73]. The association of cryptococcosis and COVID-19 is unclear, although the use of corticosteroids or immunomodulators may affect the reactivation of *Cryptococcus* spores, which can remain dormant for a long period of time [74,75].

Currently, few cases of *Cryptococcus* spp. infection have been reported in patients with COVID-19. A recent study showed 13 cases of COVID-19-associated *Cryptococcus* infection. The mean age of the patients was 73 years and they used mechanical ventilation. In total, 92.3% of patients received corticosteroids [76].

The Infectious Disease Society of America (IDSA) currently recommends the use of dexamethasone for the treatment of COVID-19, which leads to a 34% reduction in 28-day mortality among hospitalized critically ill patients compared to those not treated with glucocorticoids [77]. Dexamethasone and Methylprednisolone, both glucocorticoids, have been shown to increase the proliferation of *C. neoformans* fungal cells, and infected animals died earlier after treatments when treated with these drugs. There was an increase in the production of pulmonary and cerebral IL-10, and a reduction in the production of IL-6, after treatments with glucocorticoids [78].

*C. laurentii* was able to cause ocular cryptococcosis in a post-COVID-19 patient. The treatment used was intravitreal voriconazole twice a week, followed by systemic fluconazole for three months. Early diagnosis and rapid intervention helped to achieve favorable anatomical and visual outcomes [79].

### 4.3. Rhodotorula mucilaginosa 

*Rhodotorula* spp. forms characteristic pink colonies [80]. *Rhodotorula mucilaginosa* fungemia was defined as a rare fungal infection according to the guidelines of the European Society for Clinical Microbiology and Infectious Disease [81]. In recent years, it has been isolated from a variety of medical devices, including flexible bronchoscopes and central venous catheter tips, due to their strong plastics capacity and high biofilm-forming capacity [82]. It also been reported to cause fungemia in patients with immunodeficiency [83].

Fungemia caused by *R. mucilaginosa* in a patient with COVID-19 was reported in Iran. The isolate showed high minimum inhibitory concentration (MIC) of all azoles and echinocandins tested, but low MIC of AmB. The patient was treated with fluconazole and showed clinical evolution [84]. Guidelines recommend avoiding echinocandin therapy for patients suspected of having these yeast infections [80]. Highly resistant to antifungal agents, the *Rhodotorula* species causes approximately 50% of non-*Candida* and *Cryptococcus* fungemia cases in patients with malignancy [85]. The overall mortality rate was approximately 12% before the pandemic [80]. The diagnosis of this mycosis can be difficult because it is a capsulated yeast and can be confused with several fungal agents; until recently, it was considered nonpathogenic, making it necessary to alert health professionals for a correct diagnosis of rhodotorulosis [86].

### 4.4. Trichosporon *spp.*

*Trichosporon* is a part of the normal flora of the human skin, vagina, and gastrointestinal tract. Out of 16 *Trichosporon* species implicated in human infections, 6 are of primary clinical relevance, including *T. asahii, T. asteroides, T. cutaneum, T. inkin, T. mucoides*, and *T. ovoides.* Others, such as *T. japonicum, T. jirovecii, T. dermatis, T. coremiiforme, T. domesticum, T. faecale, T. loubieri, T. dohaense,* and *T. mycotoxinivorans*, rarely cause human disease [87]. This fungus can cause a wide spectrum of diseases, from superficial infection in immunocompetent individuals, to fungemia and invasive trichosporonosis in immunocompromised hosts [87].

During the SARS-CoV-2 pandemic, there were co-infections with *Trichosporon asahii* in North America [88], South America [89,90], Europe [91], and the Middle East [92], presenting as nosocomial pneumonia [91], fungemia [89,92], or urinary tract infection [88]. All patients in this study received broad-spectrum antibiotic therapy and systemic corticosteroids, and 55.5% received antifungals previously. The mortality rate was 77.7% [90]. *T. dohaense* caused brain abscess associated with COVID-19, a rare infection not previously described, in the central nervous system of a diabetic patient aged 55 years [93].

Currently, there are no established MIC cutoffs for antifungal drugs against *Trichosporon* spp. They are known to be inherently resistant to echinocandins and demonstrate variable MICs for fluconazole. Although most *Trichosporon* spp. show high MICs for AmB (1μg/mL), *T. dohaense* consistently demonstrated low MICs. Among the more recent triazoles, voriconazole (0.125 1μg/mL) and posaconazole (0.25 1μg/mL) exhibit potent in vitro activity against *T. dohaense* [93,94]. Invasive trichosporonosis carries a high mortality rate (40–60%) despite antifungal therapy [95]. 

For diagnosis, direct microscopy of specimens and culture is required to produce an isolate for susceptibility testing and identification of species of isolates. Identification by phenotypic methods and by MALDI-TOF MS are both moderately supported, with strong support for molecular basis identification. For susceptibility testing, particularly for *Trichosporon* spp., clinical cutoffs and epidemiological cutoff values for all antifungal drugs are scarce [96].

### 4.5. Pneumocystis jirovecii

Due to the focus on COVID-19, *Pneumocystis jirovecii* diagnoses may be missed at first [96] presentation, mainly due to similarity in radiological features. The approach to diagnosing it in patients with COVID-19, like that of other clinical settings and populations, uses clinical findings, radiographic images, and laboratory tests [97]. Traditionally, the definitive diagnosis of *Pneumocystis* pneumonia (PCP) has depended on microscopic visualization of *P. jirovecii* in respiratory tract specimens [97]. 

Many COVID-19 patients, especially those in the intensive care unit (ICU), can develop lymphopenia and acute respiratory distress syndrome, requiring adjuvant corticosteroids; these are established risks for *P. jirovecii* pneumonia [98,99]. The first report describes an immunocompetent patient with COVID-19 with a positive qualitative PCR test for *P. jirovecii* performed on a tracheal aspirate sample [97,100]. Additional cases of PCP/COVID-19 co-infections have been reported, mainly in patients infected with the human immunodeficiency virus (HIV) [101]. In 20 PCP cases reported by October 2021, 30% of patients also reported underlying HIV infection [102]. Mortality rates did not vary between HIV and non-HIV groups (43% vs. 40%, respectively), with an overall mortality rate of 41.6% [103].

In a study of consecutive COVID-19 patients admitted to a French ICU, *P. jirovecii* DNA was detected by real-time quantitative PCR in sputum, tracheal aspirate, and bronchoalveolar lavage samples [104]. As far as antifungal therapy is concerned, the approach should be like that for patients without COVID-19. Trimethoprim/sulfamethoxazole remains the preferred initial therapy for most patients [105]. 

### 4.6. Saccharomyces cerevisiae 

The genus *Saccharomyces* is a well-studied group of yeasts, and its most famous representative is *Saccharomyces cerevisiae*, which is widely used in baking, ethanol, and wine production, in addition to being used in the pharmaceutical industry. In humans, the genus *Saccharomyces* may be present as a commensal of the gastrointestinal, respiratory, and urinary mucosa [106]. Although rare, it can cause many types of deep infections [107]. The presence of *Saccharomyces* in sterile biological fluids indicates a gastrointestinal leak or high concentration of the fungus, caused by gastrointestinal injury (diarrhea, ulceration, intestinal surgery, hemodialysis, chemotherapy, and/or ischemia) [108].

A recent study in Brazil demonstrated a case of a COVID-19 patient who received *Saccharomyces* supplementation due to diarrhea and developed fungemia [107]. Another work reported two cases of critically patients who had to be admitted to the ICU due to COVID-19, also received *S. cerevisiae* supplementation, and subsequently developed bloodstream infections [109].

Severe acute respiratory syndrome coronavirus 2 (SARS-CoV-2) infection is an aggravating factor, as it can invade enterocytes, cause dysbiosis, and induce gastrointestinal symptoms, further damaging the intestinal mucosa [107]. There is no consensus on the treatment of *Saccharomyces* infection, favoring the use of fluconazole, voriconazole, flucytosine, AmB, and even the association of AmB with fluconazole [108].

Although more data and observations are needed, the occurrence of the reported cases of two patients with severe COVID-19 with long periods of ICU stay and concomitant *S. cerevisiae* bloodstream infection indicates the need for cautious use of the relevant probiotic preparations in patients with COVID-19.

### 4.7. Malassezia *spp.*

The *Pityrosporum folliculitis* condition is characterized by the overgrowth of yeasts from the *Malassezia* family, causing inflammation in the hair follicles that occurs due to a modification of the skin flora [110]. In Japan, three obese men with SARS-CoV-2 co-infection, aged 39, 46, and 52 years, were diagnosed by histological examination with chronic suppurative folliculitis with *Malassezia* species. The first two possessed no other comorbidities and the third was type 2 diabetic; the three used systemic corticosteroids [111]. 

## 5. Dimorphic Fungal Yeast Infection

Dimorphic fungal infections possess yeasts of several genera in their parasitic form that are responsible for the development of potentially fatal diseases, especially in patients with COVID-19 [112].

### 5.1. Histoplasma capsulatum

Histoplasmosis is a systemic mycosis, highly endemic in certain regions of America and Asia, including Brazil and India. It is caused by a dimorphic fungus, predominantly *Histoplasma capsulatum*, which occurs predominantly in soil containing large amounts of bird or bat droppings. Infection occurs through inhalation of fungal microconidia after disturbance of these environmental sources [113].

Histoplasmosis co-infections with COVID-19 affected two women and one man in the range between 36 and 43 years of age. Predisposing factors included HIV. The diagnosis in these cases was performed through histology (Grocott-Gomori’s methenamine silver (GMS), Wright, and Giemsa), blood culture, and antigen detection in urine and serum. Antifungal treatment in COVID-19 patients with histoplasmosis was itraconazole and AmB deoxycholate, which noticeably improved patients’ clinical state [114,115,116].

A case of disseminated histoplasmosis and COVID-19 infection in a 57-year-old kidney transplant recipient was reported in Argentina. The patient was initially treated as having bacterial pneumonia and then tuberculosis. One month later, histoplasmosis was diagnosed in sputum, skin, and oral lesions. The patient was hospitalized and started treatment with intravenous liposomal AmB. Finally, the progression was favorable, and she was discharged after five days of oral treatment with itraconazole for histoplasmosis [115].

In Brazil, two male patients aged 20 and 32 years with COVID-19 and negative anti-HIV serology were diagnosed with serology for histoplasmosis and treated with itraconazole. Both patients showed good adherence to treatment and progressed well [116].

A 33-year-old Japanese woman who had travelled through El Salvador and Brazil returned to her country after complications from COVID-19. After further evaluation, she underwent video-assisted thoracoscopic surgery. The pathological image showed a yeast-like fungus and epithelioid granuloma with necrosis in the center on Grocott’s silver methenamine stain, diagnosing *Histoplasma* by polymerase chain reaction; oral itraconazole was started with good evolution [117].

In the United States, a 50-year-old man with a history of mild intermittent asthma—well-controlled with albuterol and fluticasone-salmeterol, and with no recent need for oral corticosteroids—had COVID-19 pneumonia treated with dexamethasone. After discharge and a worsening condition, the patient tested positive for *Histoplasma* by galactomannan antigen test. He was treated with AmB followed by oral fluconazole [118]. A 61-year-old man in Texas with a history of obesity, benign essential hypertension, and controlled type 2 *diabetes mellitus*, who was not vaccinated for COVID-19, became infected and developed the disease. After complications on admission, he had positive serum and urinary *Histoplasma* antigen, and was treated with AmB followed by itraconazole [119].

### 5.2. Coccidioides *spp.*

Coccidioidomycosis is an infectious disease caused by *Coccidioides immitis*, mainly found in California, Washington State, Arizona, and Utah. *C. posadasii* is mostly found in Arizona, New Mexico, Texas, Northern Mexico, and parts of Central (Guatemala and Honduras) and South America (Northern and Central Argentina, Bolivia, Colombia, Northeastern Brazil, Paraguay, and Venezuela) [120].

A recent work-study systematically examined the risk for co-infections among construction and agricultural workers, incarcerated persons, black and Latino populations, and persons living in high-dust areas. Common risk factors for co-infection are age, diabetes, immunosuppression, racial or ethnic minority status, and smoking [120].

All the cases found with co-infection between *Coccidioides* spp. and COVID-19 were in the United States. In a single medical center within the Coccidioidal-endemic area in the United States, 13 patients met the criteria for having coccidioidomycosis simultaneously with, or following, COVID-19 infection. Their mean age was 55 years (ranging from 17 to 82) and most were male [121].

At another hospital, a previously healthy 23-year-old African American working onsite tested positive for SARS-CoV-2, then tested positive for anti-coccidiosis IgG and IgM [122]. In California, a 48-year-old Hispanic woman with systolic heart failure tested positive for COVID-19 and three days later returned to the hospital and tested positive for *Coccidioides* [123]. In the same state, a 65-year-old man, born in Mexico, with a history of poorly controlled insulin-dependent diabetes, was diagnosed with SARS-CoV-2. After using dexamethasone, and repeated hospital visits without diagnosis, the patient began liposomal AmB. Posthumously, *C. immitis* and *C. albicans* were isolated in tissue cultures from their punch biopsies [124].

In Texas, a 52-year-old Hispanic male with obesity and *diabetes mellitus* tested positive for SARS-CoV-2, received dexamethasone, and after many days in hospital with complications, was diagnosed with *C. posadasii* by MALDI-TOF MS; however, the patient died [125]. In Arizona, a 48-year-old woman with a past medical history of smoking, gastroesophageal reflux disease, urinary incontinence, and bronchitis, presented in our outpatient clinic with a two-week dry cough and diffuse pruritic nodular rash on the back, arms, and lower extremities. The patient tested positive for COVID-19 after the symptom finding was positive for both IgG and IgM coccidioidal antibodies in serum [126].

### 5.3. Paracoccidioides *spp.*

Paracoccidioidomycosis is a systemic fungal disease that occurs in Latin America and is more prevalent in South America [127]. Infection begins in the lungs after inhalation of *Paracoccidioides brasiliensis, P. americana, P. restrepiensis, P. venezuelensis,* and *P. lutzii* present on the ground, and is endemic to certain geographic regions of Central and South America [128].

The only reported case of co-infection with SARS-CoV-2 was in Brazil: a male patient, 19 years old, with the presence of multiple fungal budding structures typical of *Paracoccidioides* spp. in silver staining. Specific serum antibodies against *Paracoccidioides* spp. Were detected in the immunodiffusion test. The patient showed improvement in the clinical case after treatment with AmB lipid complex [127].

### 5.4. Geotrichum klebahnii 

The genus *Geotrichum* encompasses several species of saprophytic yeasts that colonize human skin, and respiratory and gastrointestinal tracts [129]. Trauma and burns are the only predisposing factors described in the literature for the development of geotrichosis in immunocompetent individuals [129]. It is a rare emerging pathogen that can cause an invasive disease, referred to as geotrichosis, in immunocompromised adult hosts [130].

A recent study demonstrated a case of *Geotrichum* spp. in an immunocompetent host with COVID-19—a middle-aged male with *diabetes mellitus*, using immunosuppressive agents during his hospitalization, including high doses of corticosteroids and tocilizumab, which may have increased susceptibility to fungal infection [129]. Bronchopulmonary geotrichosis should be considered in patients with severe COVID-19, particularly those who have an underlying immunocompromised state and those receiving corticosteroids or other immunosuppressive agents [129].

## 6. Concluding Remarks

Considered harmless until a few years ago, fungi have gained increasing importance in medicine due to the increase in the number of immunosuppressed patients caused by diseases or medications. In addition, due to aggression and environmental changes, it has been hypothesized that fungi have become more virulent and resistant to antifungal agents. Early diagnosis is important to individualize treatment strategies for each clinical situation. The mycological clinical picture is further worsened due to the difficulties and delay in making a definitive diagnosis of fungal diseases. In this sense, health professionals must think more about fungal diseases so that they are not neglected. Fungal diseases are difficult to diagnose and there is still a great amount of underreporting. This review is intended to raise awareness of the importance of early detection and treatment of fungal diseases in COVID-19 patients, to reduce the risk of mortality. Due importance has not been granted to the diagnosis and treatment of fungal infections, and many individuals are already suffering, and will suffer, the consequences of this neglect. In many cases, such as those mentioned in this review, the suspicion of a fungal infection is only raised as a last option, when there is no longer any way to reverse the patient’s clinical condition. This delay—on the part of clinical entities—in ordering laboratory tests for the diagnosis of mycoses is linked to the lack of knowledge about the recurrence and emergence of new fungal infections. The importance of asking for a laboratory test at the right time is even more important because, as some fungal species take time to grow, time becomes an ally for the correct diagnosis. In addition, many fungi have similar morphology, and this also makes diagnosis difficult, since it needs trained human material. Regarding treatment, a new class of drugs has not been launched on the market for decades. This highlights the lack of interest in research and development of new antifungals. Added to the recent facts of fungal resistance to some drugs, the misuse of antifungals, and increasingly immunosuppressed patients, either pathologically or through drugs, the clinical scenario that we may soon encounter is alarming, given that new and old fungal infections are already increasingly emerging.

## Data Availability

Not applicable.

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
