# Peer review of "Clinical Challenges of Emerging and Re-Emerging Yeast Infections in the Context of the COVID-19 Pandemic"

_microorganisms, 2022, doi:10.3390/microorganisms10112223_

Round 1
Reviewer 1 Report (Previous Reviewer 2)
The review "Clinical challenges of emerging and re-emerging yeast infections in the context of the COVID-19 pandemic" lays out a comprehensive overview of rare yeast infections complicating COVID-19. The review compiles interesting data and gives some food for thought. However, several points need to be amended before this manuscript is fit for publication.
L31 "geological Eras" shouldn't this be over the geological timescale?
L32 "fungal resistance" what do the authors mean by resistance. This is normally in the context of drug but here I don't see the relevance?
Section 2. The rationale for talking about corticosteroids is not clear. The review is in the context of COVID-19. Before stating more specific details about corticosteroids the authors should talk about its use in COVID-19 treatment. It is lacking much detail - how many people are on treatment, where, what patient groups etc.
Section 4.1 Authors should try and find incidence rates in COVID-19, mortality, treatment failure, resistance etc to add to this section. More specific numbers are required.
Section 5 is missing Blastomyces, there is some interesting data on COVID-19.
Author Response
REVIEWER 1:
The review "Clinical challenges of emerging and re-emerging yeast infections in the context of the COVID-19 pandemic" lays out a comprehensive overview of rare yeast infections complicating COVID-19. The review compiles interesting data and gives some food for thought. However, several points need to be amended before this manuscript is fit for publication.
COMMENT 1: L31 "geological Eras" shouldn't this be over the geological timescale?
AUTHOR RESPONSE & ACTION TAKEN: We modified this sentence.
COMMENT 2: L32 "fungal resistance" what do the authors mean by resistance. This is normally in the context of drug but here I don't see the relevance?
AUTHOR RESPONSE & ACTION TAKEN: We modified this sentence and removed the term resistance.
COMMENT 3: Section 2. The rationale for talking about corticosteroids is not clear. The review is in the context of COVID-19. Before stating more specific details about corticosteroids the authors should talk about its use in COVID-19 treatment. It is lacking much detail - how many people are on treatment, where, what patient groups etc.
AUTHOR RESPONSE & ACTION TAKEN: We tried to improve the text and contextualize the use of corticosteroids and the link with covid-19.
COMMENT 4: Section 4.1 Authors should try and find incidence rates in COVID-19, mortality, treatment failure, resistance etc to add to this section. More specific numbers are required.
AUTHOR RESPONSE & ACTION TAKEN: We modified this sentence and added the information
COMMENT 5: Section 5 is missing Blastomyces, there is some interesting data on COVID-19.
AUTHOR RESPONSE & ACTION TAKEN: Unfortunately, we were unable to access these works. The data found were from a patient with blastomycosis who later had covid-19. Sorry for not being able to add more information.
Reviewer 2 Report (New Reviewer)
The phrase on line 394-395 does not make complete sense.
I suggest to report cases of SARS-CoV-2 association with Candida auris and Candida non-albicans candidiases, as reported at other fungal diseases cited in the paper.
Author Response
REVIEWER 2: The phrase on line 394-395 does not make complete sense.I suggest to report cases of SARS-CoV-2 association with Candida auris and Candida non-albicans candidiases, as reported at other fungal diseases cited in the paper.
AUTHOR RESPONSE & ACTION TAKEN: We tried to improve the text and added this information
Reviewer 3 Report (New Reviewer)
The manuscript addresses a relevant subject, but currently it seems that lacks structure and objective. One of the main aspects (SARS-CoV-2 infection) is barely incorporated into the literature revision. The section about non-albicans Candida species adds not much to the discussion, as few cases have been reported. In a similar line, the glucocorticoids and introduction sections layout a different aim pursued by the manuscript. I honestly think that all the immunosuppression-related material is out of the scope of a revision of this type. There are several reports of coinfection with COVID-19 and Aspergillus or Mucor, for example, which are not part of the manuscript. Tables and Figures should also be included to complement the text.
Author Response
AUTHOR RESPONSE & ACTION TAKEN: Part of the information requested by the reviewer has been modified. However, due to the lack of details about what is being requested, we were unable to make further changes. Our review is about yeast infections. Therefore, Aspergillus and Mucor are not included.
Round 2
Reviewer 3 Report (New Reviewer)
I thank the authors for the revised version, but honestly, I do not see any improvement considering my previous concerns that still stand. I still see two subjects disconnected fungal infections and COVID-19 that barely touch each other throughout the text. Again, I do not understand the bias towards only immunosuppressed patients, most of the cases of COVID-19 are reported in an immunocompetent population. The authors may argue that some, but not all COVID-19 patients are treated with glucocorticoids paving the way to temporal immunosuppression. Still, in the strict sense, only a few cases it has been documented immunosuppression in this group of patients. Once again, I recommend focusing the review manuscript on COVID-19 patients with fungal infections, ditching out the immunosuppression, and broadening the scope to filament fungi. As a side note, Mucor also undergoes dimorphism and it is not included in the section dealing with these kinds of fungal pathogens.
Author Response
REVIEWER 3:
I thank the authors for the revised version, but honestly, I do not see any improvement considering my previous concerns that still stand. I still see two subjects disconnected fungal infections and COVID-19 that barely touch each other throughout the text. Again, I do not understand the bias towards only immunosuppressed patients, most of the cases of COVID-19 are reported in an immunocompetent population. The authors may argue that some, but not all COVID-19 patients are treated with glucocorticoids paving the way to temporal immunosuppression. Still, in the strict sense, only a few cases it has been documented immunosuppression in this group of patients. Once again, I recommend focusing the review manuscript on COVID-19 patients with fungal infections, ditching out the immunosuppression, and broadening the scope to filament fungi. As a side note, Mucor also undergoes dimorphism and it is not included in the section dealing with these kinds of fungal pathogens.
AUTHOR RESPONSE & ACTION TAKEN: Part of the information requested by the reviewer has been modified in the two first versions. Our review has the aim proposed in the initial invitation and we do not agree to change the focus now. We do not think it is appropriate to switch to a type of review that already has several in the literature.
Corticosteroids exert their anti-inflammatory and immunosuppressive activity by interfering with different stages of immune system regulation. Because of their immunosuppressive and anti-inflammatory properties, corticosteroids have been prescribed to infected patients with Severe Acute Respiratory Syndrome (SARS) with severe pulmonary condition, with the aim of reducing the inflammatory process associated with the exacerbated production of cytokines, pulmonary edema, and alveolar damage; consequently improving hypoxia and reducing the risk of respiratory failure. Data in the literature provide compelling evidence on the effects of dexamethasone and methylprednisolone on the biology of fungal infections and may have important implications for future clinical treatments, warning of the risks of using these glucocorticoids. The use of corticosteroids is a widely studied risk factor for the development of invasive aspergillosis in critically ill patients. de León-Borrás et al. showed that patients who receive corticosteroids have a 3.33-fold increased risk to develop an IFI in comparison to other patients that did not receive steroidal treatment. Temporary immunosuppression is sufficient for high risk of fungal infections and the alert must be made.
On the other hand, we appreciate the reviewer's opinion on how the article should be done, but we would like to keep the focus agreed in the initial invitation.
This manuscript is a resubmission of an earlier submission. The following is a list of the peer review reports and author responses from that submission.
Round 1
Reviewer 1 Report
The authors in this study proposed a review to survey emerging and re-emerging yeast infections in the current clinical context. The authors reported the immense clinical challenge for the rapid and correct diagnosis and treatment of systemic mycoses caused by yeasts due to the terrible increase in cases in the current context of COVID-19.
However, the authors have some problems in methodology applied and design this study. Being the main problem in this study was not define the time for inclusion of studies in literature search. Other problem is restriction in few yeast infections.
I suggest review methodology and including more microorganism of fungi kingdom.
Briefly, I suggest that the authors carry out review the design of study for a new evaluation of this study in future.
Below are included some suggestions or modifications that may improve the content of the manuscript in comments to the Author.
Comments to the Authors:
Introduction
The authors should review the text and all references. I suggest include references that reported some data about clinical challenges in treatment of infections by fungi.
Methods
I suggest include the session “methods”.
In this section the authors must review the strategies of literature search with the inclusion of time.
In text where were reported results, the authors must review this text and include more data.
In topic 5.1. the authors must review the title this topic because the scientific names of species should be italicized.
The authors should review the section “Concluding Remarks”. In this section I suggest increase the text and discussion about this important thematic.
Reviewer 2 Report
The manuscript "microorganisms-1861769 Clinical challenges of emerging and re-emerging yeast infections in the context of the COVID-19 pandemic" surveys rare and emerging yeast infections in a COVID-19 setting. Clinical implications and challenges are described for these particularly difficult to treat patients. The review presents some interesting findings on a generally overlooked subject. However, in its current format this review lacks structure, omits many scientific literature and is overall not complete. Many of the references used are not supporting statements provided and for a review, primary literature needs to be used more. It would require a significant overhaul to be ready for publication.
General comments:
It is unclear why section 2.1 and 2.2 link to COVID and fungal infections. There is plenty of literature on the effect of Corticosteroids and Cyclosporin on fungal infections and its use in COVID. Much more information is needed and why specifically these two drugs (classes) have been chosen.
Section 3 is generally immunology during COVID-19 but the link to fungal infections is unclear. Requires much more information why this directly links to fungal infections, and more specifically of emerging and re-emerging yeasts
The whole classification of dimorphic fungi is wrong, dimorphics have multiple morphological stages, not only temperature switch at the temperature mentioned. It is way more complex. Also, blastomycosis is omitted?
Many sections are purely descriptions of case reports. For a review I would expect drawing parallels, explaining clinical challenges of diagnosis and treatment.
- L27: "Fungi Kingdom" The Kingdom "Fungi" or the kingdom of fungi
- L30: "adapted their pathogenic repertoire primarily" this is an overgeneralisation and is theoretical + unproven.
- L32: "and attacking" attacking would imply an active component. The authors have just said fungi are generally accidental pathogens.
- L36: "parasitise" not correct for fungi
- L52-54: needs reference that show climate change adaptations and adaptation to environmental pressures. Casadevall et al, van Rhijn et al have looked at climate change.
- L64: "increase the incidence of infectious diseases" this statement needs a ref if this is claimed.
- The section 2.1 contains abbreviations which are only used once. No need to abbreviate if only used once.
- L110-111: "in the immunohistopathological" do the authors mean, variability in immunohistological presentation?
- L114: "like" either use "similar" or "within the same family"
- L124: "horizon" I do not understand the use of this word here
- L129: "parasite" viral-host relationship
- L150-152: needs better refs. This does not support the statement
- L152-153: "it is the most important fungal genus in the medical field" is subjective and definitely not true.
- L153: Ref 41 is a C. auris review and does not support the statement
- L154: "the main species found" found where? Not everywhere
- L156-157: These species had a reclassification. Use the new names here.
- L162: Ref 46,47 are case reports. Does not support a general statement about risk factors in larger group of patients
- L167: Ref 51 is not primary data. Find better ref
- L173: "potentiated" what do the authors mean here?
- L174: needs a ref.
- L179: "being detachable" unclear what authors mean
- L182-183: needs refs. There have been multiple outbreaks in C. auris in Florida, Mexico and lots of info is available
- L186: Ref 58 is about Mucor. Not correct ref here
- L187: "allowed many fungal species to acquire resistance" Is not shown, there is correlation between bad stewardship and resistance but acquiring directly is not shown in these refs.
- L188-194: Why this section? I don't understand what the point is of describing this whole report. C glabrata is intrinsically resistant to many antifungals so what are the authors trying to say?
- L195-197: needs refs
- L196: "micafungin and anidulafungin" not and, but OR
- L198: "azole toxicity" ampB toxicity even more, include all antifungals. This needs a ref.
- L199: Ref 60, 61 do not support this statement
-L200-207: this is a small report, I do not understand the point of describing it all in such detail.
- L208-210: "Many deaths" but only one case reports is mentioned. If this general statement is made, better literature needs to be found.
- L217-218: needs a ref
- L221: ref 64 is an aspergillosis article, not correct reference
- L227: "most reliable method" do authors mean accurate, sensitive? Reliable is not correct term here
- L229 ITS do not encode rRNA subunits, they are introns so do not encode anything.
- Section 4.1 Misses information on how often superinfection of Candida and COVID occur, cause of death, mortality etc.
- L244: "Seven species' not correct and needs a ref
- L285: "major clinical relevance" they are most definitely not, they are rare
- L286-288: These had a reclassification, use correct species name. See Liu et al 2015
- L295: "mean age of 66.4 years" This is just of one study, can't make generalisation like this.
- L298: "rare species" it is not a rare species, but rare infection causing.
- L300: no MIC cut-offs for Trichosporon. There are many articles reporting MIC cut-offs.
- L301-302: needs a ref
- L305-306: "can't make generalisation based on case report
- L313: ref 92 is not correct
- L314: "ICU, have" use "can"
- L316: Ref 93 is not correct for this statement
- L318: Ref 94 is not correct
- L320: Ref 93 is not correct
- L321: Ref 95 is not correct
- L330: Ref 99 is not correct
- L335: coloniser. It is a commensal
- L335: Ref 101 only shows one type, not correct
- L336: Ref 100 is not correct here
- L354: Geotrichum is also a dimorphic fungus
- L356: "is found in nature" aren't all fungi?
- L359: "that causes" use "that can cause"
- L368-372: This is about coccidioidomycosis?
- L373: Malassezia is a dimorphic fungus so your classification does not hold up.
- L374-379: This section does not describe any clinical challenges.
- L386: causes by Histoplasma capsulatum. There are more species causing it.
- L392: "Staining methods" histology?
- L400: "Patient evolution was favourable" do authors mean progression?
- L430-431: This sentence does not make sense?
- L432-453: Describes some case reports. For a review I would expect drawing some parallels and explaining the clinical challenge.
- L468-469: "more virulent and resistant to antifungal agents" This has not been proven. Can't claim this.